# Effect of Reduced Feedback Frequencies on Motor Learning in a Postural Control Task in Young Adults

**DOI:** 10.3390/s24051404

**Published:** 2024-02-22

**Authors:** Adrià Marco-Ahulló, Israel Villarrasa-Sapiña, Jorge Romero-Martínez, Gonzalo Monfort-Torres, Jose Luis Toca-Herrera, Xavier García-Massó

**Affiliations:** 1Departamento de Neuropsicobiología, Metodología y Psicología Social, Universidad Católica de Valencia “San Vicente Mártir”, 46001 València, Spain; adria.marco@ucv.es; 2Departament d’Educació Física i Esportiva, Universitat de València, 46010 València, Spain; israel.villarrasa@uv.es; 3Departament de Didàctica de l’Educació Física, Artística i Música, Universitat de València, 46022 València, Spain; jorge.romero-martinez@uv.es (J.R.-M.); gmonfort@florida-uni.es (G.M.-T.); 4Unidad de Educación, Florida Universitaria, 46470 València, Spain; 5Institute of Biophysics, Department of Nanobiotechnology, University of Natural Resources and Life Sciences, 1180 Vienna, Austria; jose.toca-herrera@boku.ac.at

**Keywords:** augmented feedback, postural control, motor learning, extrinsic feedback

## Abstract

The effects of the use of reduced feedback frequencies on motor learning remain controversial in the scientific literature. At present, there is still controversy about the guidance hypothesis, with some works supporting it and others contradicting it. To shed light on this topic, an experiment was conducted with four groups, each with different feedback frequencies (0%, 33%, 67%, and 100%), which were evaluated three times (pre-test, post-test, and retention) during a postural control task. In addition, we tested whether there was a transfer in performance to another similar task involving postural control. As a result, only the 67% feedback group showed an improvement in their task performance in the post-test and retention evaluations. Nevertheless, neither group showed differences in motor transfer performance compared to another postural control task. In conclusion, the findings of this paper corroborate the hypothesis of guidance and suggest that the use of a reduced frequency of 67% is a better option for improving motor learning than options that offer feedback at a lower frequency, at all trials or not at all.

## 1. Introduction

Feedback is used to optimize motor learning in different age groups [1,2,3]. In essence, motor learning feedback is based on providing information about the performance of an action or task to the practitioner (i.e., the person who performs the action/task), which is used as a basis for improvement [4].

In the scientific literature, different types of feedback can be found [1,5,6]. Generally, feedback can be classified into two main groups according to the source of the information: intrinsic and extrinsic feedback. On the one hand, intrinsic feedback is the information that a practitioner receives from himself/herself about the action or task that he/she is engaging in. On the other hand, extrinsic or augmented feedback can be defined as the information that the practitioner receives from a source external to him/her (e.g., something around him/her or another human) [7,8]. This feedback seems to be the one that has received the most attention in related research, because science seems to agree that extrinsic/augmented feedback provides better performance than internal feedback for motor learning [9]. Therefore, researchers are focusing their studies on extrinsic feedback.

Extrinsic feedback has various subtypes. First, according to the sensory pathway through which the learner receives the information, feedback can be visual (the most sensorial feedback studied [1]), auditory, or haptic (i.e., all feedback that is not visual or auditory) [1,10]. Second, according to the type of information that is given, information can concern the knowledge of results or the knowledge of performance [11]. Third, according to the moment at which information is transmitted to the subject, feedback can be terminal (e.g., the information is received after action) or concurrent (e.g., the information is received during action) [2]. Finally, other classifications use the frequency with which feedback is provided as a criterion. In this regard, feedback can be continuous (i.e., providing feedback during all attempts or for the entire duration of the task, such as 100% of feedback) or intermittent (i.e., providing feedback on some attempts or partially during the task). It is often examined using frequency percentages, e.g., giving the subject feedback during 100% or 67% of the time he/she is performing an action/task [5,12,13]. It should be noted that sensors used for motor learning are crucial for acquiring and providing useful information to practitioners, and are particularly useful for adjusting the frequency of feedback provided. In this sense, low-cost portable sensors are tools that have experienced an exponential boom in recent years, and their application in the field of motor learning is highly interesting [14].

If we focus on the types of frequency at which feedback is provided, we must mention the guidance hypothesis. This hypothesis suggests that an excess of extrinsic feedback can be detrimental to motor learning [15]. This hypothesis, proposed by Salmoni [16], indicates that extrinsic feedback is a great tool for promoting motor learning; however, when feedback is presented too frequently, the subject may overlook important intrinsic processing mechanisms and become overly dependent on the external source at the expense of their intrinsic feedback [17,18]. This can lead to worsening or slower progress of motor learning.

From a neurological point of view, there are studies that support this hypothesis. For example, Ronsse et al. [19] conducted a study in which each of the two groups participating in the experiment was assigned to one type of feedback intervention (visual or auditory). As a result, using functional magnetic resonance imaging, they found that the group that received visual feedback showed increased neural activity in specific sensory areas during practice, and even found traces of brain activity in these areas in the absence of feedback, supporting a greater reliance on this tool. In contrast, users of auditory feedback showed reduced neural activity, which was associated with less reliance on feedback. The authors of the aforementioned paper suggested that the visual information received by the subjects becomes an integral part of the sensorimotor representation, which may lead to insufficient attention to proprioceptive sources of information.

During the last three decades, although the argument for the guidance hypothesis appears to be well supported, it seems that not all research results related to feedback frequencies are in the same direction. While some articles obtained results in line with the guidance hypothesis, indicating that the use of a reduced frequency is more appropriate for adaptation or motor learning [20,21,22], other studies reported findings indicating no difference between the use of different feedback frequencies [23,24,25,26]. A recent meta-analysis addressed the issue of the effect of reduced feedback frequency on motor learning and concluded that additional research is needed in this area to confirm the guidance hypothesis, as most of the studies conducted to date have too little statistical power (three out of four studies) and may be biased towards more conclusive results [7]. This conclusion can be drawn from recent reviews or even from different populations [1,13]. It should also be noted that the studies that have been carried out have generally used few feedback frequencies, so it is necessary to carry out studies using a wider range of frequencies. In this way, we can determine not only whether the use of reduced feedback is a better or worse option than providing feedback on all or no trials during training, but also whether the use of a high feedback frequency is more or less effective for motor learning than a low one.

An important concept in motor learning is transfer. This refers to the use of the learning of a motor skill acquired in the practice of a particular task to achieve improvements in another similar task or in other environments [27,28]. However, more attention needs to be paid to investigating the effects of motor learning interventions based on providing different frequencies of feedback on transfer.

For all the aforementioned factors, although there is a considerable amount of research on the effects of feedback on motor learning, further studies are needed to draw conclusions with sufficient statistical power regarding the use of different types of feedback in various situations, scenarios, and skills. For this reason, the main aim of the present work was to verify the guidance hypothesis by analyzing the effects of different concurrent visual feedback frequencies (i.e., 0%, 33%, 67%, and 100%) on performance in a postural control task. In addition, the analysis of the transfer of learning to another postural control task is considered a secondary aim.

## 2. Materials and Methods

### 2.1. Participants

The sample in this study was composed of 60 young adults (between 18 and 35 years old). A preliminary sample size calculation was performed using G*Power 3.1 (University of Düsseldorf, Düsseldorf, Germany) and a Cohen’s effect size of d = 1.3 (based on data published previously [29]). The level of significance was set at 0.05, and the statistical power was set at 0.9. The results of this analysis reported a sample size of 14 subjects in each group.

The participants were divided randomly into four different groups: (i) the control group (CG), (ii) the 100% visual feedback group, (iii) the 67% visual feedback group, and (iv) the 33% visual feedback group. The inclusion criteria were (i) aged between 18 and 35 years (both inclusive), (ii) had no injuries in the last 6 months, and (iii) had no neurological or musculoskeletal disorders that could affect balance control. All characteristics are shown in Table 1.

The study had previously been assessed by the Institutional Review Board of our University (code H14879747058), and written informed consent was obtained from the participants.

### 2.2. Procedure

First, the subjects were invited to come individually to the research laboratory for two consecutive days. Once the initial contact and anthropometric measurements had been made, the procedure was explained to the subject. During the first experimental session, the subject completed a pre-test, training session, and post-test. In the second session, 24 h later, participants performed a retention test. All the tests (i.e., pre-test, post-test, and retention) included the same tasks and protocols.

#### 2.2.1. Assessments

As described above, in the pre-test, post-test, and retention tests, all the subjects completed the same protocol independently of their training group. The participants performed three 30 s trials of a postural control task involving instability in the AP direction and three other 30 s trials of a postural control task involving instability in the ML direction. The second test was used to verify the potential transfer of learning to other similar postural control tasks.

These tasks involved standing on a seesaw with instability in the anterior–posterior (AP) or medio-lateral (ML) direction (radius of the seesaw base = 486 mm), which participants had to maintain as horizontally as possible (equilibrium position) for 30 s (Figure 1). During the execution of the task, the subjects had to look at a fixed point approximately 5 cm in diameter, which was projected on a computer monitor placed at eye height at a distance of 1.5 m. The feet were placed shoulder-width apart with the toes pointing forward. A reference was placed on the platform so that participants placed their feet in the same position for all trials. The arms were held at the sides of the body in a relaxed position. Subjects were allowed to rest for 30–60 s between trials. No feedback was provided to participants during the assessments.

#### 2.2.2. Training Protocol

The training protocol was administered immediately after the pre-test and before the post-test. The test consisted of twelve 30 s trials in which participants performed the same task with instability in the AP direction, as described in the assessment. Thus, the participants trained only on the AP instability tasks and not on the ML instability task.

During training, feedback was provided according to the group in which the subjects were assigned. The CG performed the training by looking at a point projected on the screen (without feedback), as in the assessments. However, the experimental groups received concurrent visual feedback during the training trials. It is important to note that each group received feedback at a different frequency. The visual feedback was provided by a screen placed in front of the subjects at the same distance as the CG. However, instead of visualizing a single point, subjects could visualize their performance on the task in real time, observing how far away they were from the optimal performance on the task. An x-y graph was projected onto screen, with the x-axis representing time and the y-axis representing the center of pressure (CoP) position in the AP direction (Figure 1). The red line symbolized the “0” point of the CoP, and the blue line represented the subject’s performance. Therefore, the participant’s goal was to stay as close to the red line as possible for as long as possible. Additionally, as mentioned above, the experimental groups received different frequencies of feedback: (i) 100% feedback group (on all trials), (ii) 67% feedback group (on 2 out of 3 trials), and (iii) 33% feedback group (on 1 out of 3 trials). For example, the participants in the 67% feedback group viewed the screen with the instantaneous representation of their CoP for two consecutive trials. On the third trial, however, participants saw only a single point and received no feedback (i.e., they received feedback on 2 out of 3 trials, representing 67% of the trials). This sequence was repeated throughout the series until 12 trials had been completed.

### 2.3. Instruments and Postural Control Measurements

For the measurement of the postural control variables, as well as for feedback provision, a Wii Balance Board was placed on the unstable platform. This device is equipped with four sensors that measure the forces exerted on it. These sensors are placed near the corners of the platform to measure the vertical force component (i.e., Fz). Using the forced-registered and torque formulas, the CoP is calculated in both the AP and ML directions. The CoP refers to the point where all these forces are concentrated on the bearing surface. Therefore, the Wii Balance Board provides spatial information about the displacement of the CoP [30]. We placed the Wii Balance Board in the center of the seesaw. With this setting, the position of the center of pressure was proportional to the angle of the seesaw.

This instrument has been validated as a force platform in different age groups [31,32,33] and has been used to measure postural control variables in different studies [20,25]. In addition, a specific validation of the ability of the Wii Balance Board to acquire seesaw inclination was performed previously [25]. As this previous manuscript did not use the same variables as did the present study, we conducted a pilot study to validate our protocol for measuring seesaw inclination using CoP data from the Wii Balance Board. An Xsens Dot gyroscope (Xsens Technologies B.V., Enschede, The Netherlands) was used as the gold standard. The correlations between the RMS and MV obtained with both devices were 0.81 and 0.95, respectively. Raw data were acquired using custom software developed in LabVIEW 2015 (LabVIEW, National Instruments, Austin, TX, USA). Data signals were recorded at a frequency of 40 Hz.

### 2.4. Data Analysis

The CoP displacement signals were digitally filtered using a Butterworth low-pass filter with a 12 Hz cut-off frequency. Task performance (both in the ML and AP directions) was assessed using the root mean square of the signal in the direction of seesaw instability. Moreover, to assess the net neuromuscular effort required to perform the task, the mean velocity (MV) in the AP and ML directions was computed depending on whether the task was performed with anteroposterior or mediolateral instability [34]. Therefore, four outcomes, the RMS and MV in AP and ML, were computed for each of the time points (i.e., pre-test, post-test, and retention). Note that the RMS and MV in the AP were calculated for the three trials in which participants performed the task with instability in the AP, whereas the RMS and MV in the ML were calculated for the three trials with instability in that direction. Since three trials were performed in each direction at each time point, the mean of the three trials was calculated. Thus, the RMS and MV in the AP, as well as the RMS and MV in the ML, were available for pre-test, post-test, and retention. The RMS and MV in the AP were considered the main variables that allowed for the assessment of motor adaptation and learning, while the RMS and MV in the ML allowed for the assessment of the transfer of learning to the same task performed in a different direction of instability.

### 2.5. Statistical Analysis

The statistical analysis was performed using SPSS software version 25 (SPSS, Inc., Chicago, IL, USA). Four mixed-model ANOVAs were carried out to check the effect of feedback frequency (four groups: control, 100% feedback, 67% feedback, and 33% feedback) during motor learning (three testing sessions: pre-test, post-test, and retention) on RMS and MV in the AP and ML. The follow-up was performed by pairwise comparisons with Bonferroni’s correction. The significance level was set at *p* < 0.05 for all the statistical analyses.

## 3. Results

First, there were no significant differences in any of the dependent variables between the groups at baseline. For RMS in AP, mixed-model ANOVA revealed an interaction effect between testing time and feedback group (F_6,112_ = 3.26; *p* = 0.005; η^2^_p_ = 0.15). Pairwise comparisons are reported in Figure 2. The 100% feedback group had an increased RMS on the AP test between post-test and retention (*p* < 0.05). Moreover, the 67% feedback group had a lower RMS in AP in the post-test and retention test than in the pre-test (*p* < 0.05). There were no statistically significant differences in the rest of the comparisons between groups or at any testing time. Finally, the main effects of testing time (F_2,112_ = 2.49; *p* = 0.09; η^2^_p_ = 0.08) and group (F_3,56_ = 0.94; *p* = 0.43; η^2^_p_ = 0.05) were not significant.

Regarding the RMS in ML, ANOVA showed no interaction effect between the feedback group and testing time (F_6,112_ = 0.53; *p* = 0.79; η^2^_p_ = 0.47). The main effects of the testing time (F_2,112_ = 0.48; *p* = 0.62; η^2^_p_ = 0.02) and group (F_3,56_ = 1.13; *p* = 0.34; η^2^_p_ = 0.06) were not significant.

Finally, a test time effect was found for MV in both AP and ML (F_2,112_ = 48.18; *p* < 0.001; η^2^_p_ = 0.08 and F_2,112_ = 32.37; *p* < 0.001; η^2^_p_ = 0.36). Post hoc analyses showed that all groups had significantly reduced MVs in the AP and ML directions at post-test and retention compared to baseline (*p* < 0.05). However, no interaction effects between group and test time were found (F_6,112_ = 1.68; *p* = 0.13; η^2^_p_ = 0.08 and F_6,112_ = 0.39; *p* = 0.88; η^2^_p_ = 0.02). The main effect of group was also not significant for AP (F_3,56_ = 1.21; *p* = 0.32; η^2^_p_ = 0.06) or ML (F_3,56_ = 0.52; *p* = 0.67; η^2^_p_ = 0.03).

## 4. Discussion

The main objective of this study was to test the guidance hypothesis during a motor learning task. This hypothesis has been supported by different studies and applied to different age ranges and motor tasks, such as soccer throw-in skill [35], the application of a target force with a dynamometer [36], or coordination patterns [37].

The results presented in this study show that reduced feedback is more effective at learning a postural task than complete feedback (i.e., a frequency of 100%). Specifically, the obtained findings suggest that the group that used a reduced feedback frequency of 67% achieved improvements in task performance in the post-test and retention compared to the pre-test in the postural control task, with instability in the AP direction. Nevertheless, the group that received feedback on all trials (100% feedback) did not improve their performance upon post-test, and even showed a worsening in retention compared to the post-test. These results are in full agreement with the guidance hypothesis and could be due to the dependence of participants on excessive feedback [17].

However, no differences in RMS were found between any of the measurement moments in the CG or in the group that used a reduced feedback frequency of 33%. Nonetheless, it is worth mentioning that the values of the 33% feedback group improved with each test, while the CG’s values worsened compared to the pre-test (see Figure 2). Thus, these results are consistent with the guiding hypothesis [16], supporting the idea that using a reduced feedback frequency is better than providing feedback on all trials. Specifically, training with concurrent visual feedback was more effective at a frequency of 67% than 33%.

On the other hand, all the groups reduced their MV in the AP direction, so although they did not improve their performance on the task, they did reduce the neuromuscular effort required to carry out the activity.

As mentioned above in the introduction, there is evidence that, from a neurological point of view, visual feedback creates a dependency on the subject’s use of this information, causing them to pay less attention to their proprioceptive information and impairing their subsequent task performance. However, several factors can influence the results of this type of research, such as the type of feedback, the number of times the subject practices the task, or the task itself. In this case, the results support that a relatively high frequency of feedback (67%) produces improvements in task performance that do not occur under other conditions (i.e., low frequency of feedback, feedback on all trials, or no feedback at all).

These results corroborate those of a previous study performed using dynamic postural control tasks. Goodwin reported that groups with reduced feedback had significantly greater motor learning than did those with no feedback or 100% feedback. However, he found no difference between the different types of reduced feedback. This difference in the results may be because he applied feedback with no constant frequency, as in our study, since he applied it via progressive modifications during the task [38].

Other previous studies conducted with similar postural control tasks have been carried out in different populations and with a variety of feedback and frequencies. Specifically, similar results were found in older adults, as reduced feedback (50%) was significantly better than 100% feedback. However, in this population, no feedback also significantly improved motor learning compared to 100% feedback. This could be due to either the population studied or because, in this study, all participants also received terminal feedback, which could alter motor learning results [39].

Marco-Ahulló et al. used the same postural control task as in the present study and provided feedback to adolescents aged 13–14 years [20]. The results obtained are in perfect agreement with those obtained in the present manuscript, as it was found that the use of a feedback frequency of 67% was the best option for improving motor adaptation that was not maintained on the retention test. However, it should be noted that Marco-Ahulló et al. did not include a group of subjects with a frequency of 33%, so we cannot know whether adolescents could have achieved better results with an even lower frequency than 67%.

As seen herein, studies analyzing reduced feedback for motor learning in postural control tasks on unstable platforms, using different frequencies of concurrent visual feedback, have concluded that reduced feedback is more effective than 100% feedback. Therefore, it can be concluded that, in this type of task, the guidance hypothesis is supported. However, it should be specified that this is confirmed when applying visual feedback, since, when applying another input channel, the result may vary.

For example, another study with an adolescent sample used auditory feedback instead of visual feedback, and the results differed from our findings [25]. In this sense, the authors showed that providing feedback on all trials was better than providing reduced feedback frequency or no feedback at all. It is also worth mentioning that, in this case, there was also no group with a reduced feedback frequency of less than 67%. Additionally, it may be that the sensory channel through which the feedback was provided was the cause of the differences in the results. Likewise, it may be that receiving information through sight may be more useful to the participant than receiving it through the auditory channel. Thus, it may be that the information received by the subjects during training with visual feedback at a reduced frequency of 67% was sufficient for the participants to be aware of their performance on the task and to be able to transfer what they had learned to the trials in which they had no feedback.

Finally, another interesting factor is the capacity to transfer motor learning to new conditions and similar tasks or variants [40]. In this case, our study showed no significant differences in the performance of the postural control task (i.e., RMS) in the ML direction between any of the measurement moments in any of the groups. This suggests that the group that received 67% feedback and showed improvement on the trained task was not able to transfer this performance to another postural control task. These results are fully in line with previously published results obtained with postural control tasks [41] and other types of feedback and transfer [42,43,44]. However, as with the main task, all the groups were able to reduce the neuromuscular resources used to perform the task (i.e., improved MV). Therefore, these results may confirm that the type of feedback itself does not seem to alter the transfer of performance from one motor task to another, even if it is similar. One of the possible explanations for this lack of transfer is the duration of the intervention. In this sense, previous works have argued that brain regions that remain active in the first learning stage are not re-engaged in the transfer task. Nevertheless, motor transfer appears to be associated with the activation of brain areas associated with late learning and storage [40,45]. Consequently, the expected results may not have been found because of the short duration of the intervention or because the transfer task was measured in the immediate period or within one day after the intervention.

As a practical application derived from the results of this study, the authors recommend providing feedback at a frequency of approximately 67% of practice trials. This overall recommendation can be generally applied to training, health, and physical education environments. Therefore, clinicians, personal trainers, and physical education teachers should consider how to plan activities to improve motor performance through coordinative activities to provide feedback to practitioners at a frequency of approximately 67%. For that purpose, low-cost sensors such as those used in this manuscript (i.e., the Wii Balance Board with four pressure sensors) can be considered.

To cover some of the limitations of the present study and to propose new lines of research, it would be desirable to carry out field research in which reduced feedback frequencies are used to increase ecological validity. Moreover, it would be interesting to address tasks in which reduced frequencies of concurrent and intermittent visual feedback are offered. In other words, if we take the design of this study as an example, the group that is assigned 67% feedback will be provided with this tool in all trials, but only in 2 out of every 3 s (i.e., in total, in 20 out of the 30 s that the task lasts). Additionally, it would be interesting to carry out work using different feedback frequencies and different feedback modalities depending on the sensory channel in different age groups. Finally, it is worth mentioning that, although it was ensured that there were no differences in the initial levels of the subjects in the task (taking into account the groups formed, BMI, and gender), there may be other individual factors that regulate the effect of the feedback.

## 5. Conclusions

In conclusion, the present study showed that using a reduced frequency of 67% was a better option for improving motor learning than using feedback at a lower frequency, on all trials, or not at all. This contribution also adds to the scientific literature as evidence of the suitability of the use of reduced feedback frequencies for improving performance in certain activities. However, as the literature points out, this area of knowledge continues to provide contradictory results regarding the use of these tools. Therefore, continuing to improve our understanding of the application of reduced feedback frequencies is necessary. This study also confirmed that the frequency of feedback applied does not affect motor transfer to other tasks, as does the frequency of feedback applied by trained individuals. Finally, a low-cost device consisting of four pressure sensors, such as the Wii Balance Board, has been shown to be useful for providing feedback in postural control tasks on a seesaw.

## Figures and Tables

**Figure 1 sensors-24-01404-f001:**
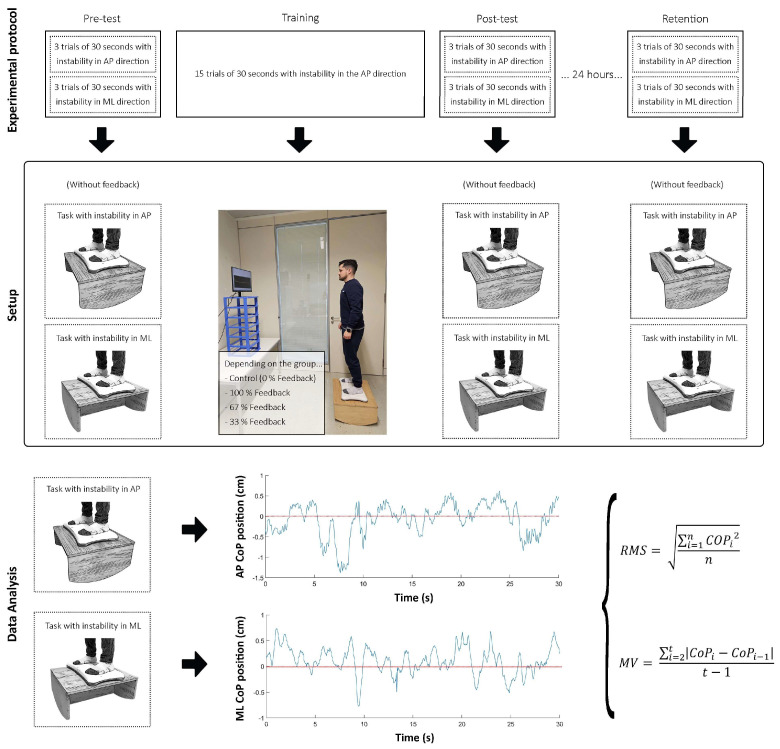
Scheme of the experimental protocol, setup of measurement and training, and data analysis.

**Figure 2 sensors-24-01404-f002:**
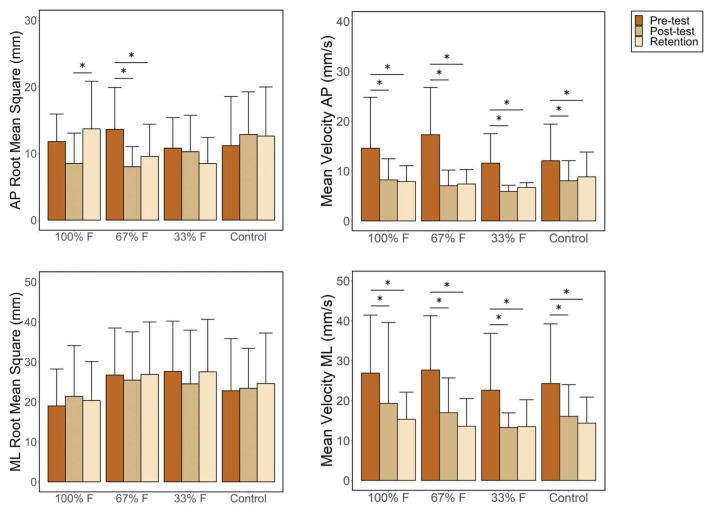
Pairwise comparisons of root mean square errors at different testing times during the AP and ML tests. The bars represent the mean, and the error bars represent the standard deviation. * indicates significant differences between testing times.

**Table 1 sensors-24-01404-t001:** Participants’ characteristics.

Group	Age (Years)	Weight (kg)	Height (cm)	BMI (kg × cm^2^)	Gender (Male/Female)
Control	21.8	67.4	173	21.8	7/8
n = 15	(3.41)	(15)	(10.3)	(3.07)	
100% Visual Feedback	23.9	72	175	23.3	11/4
n = 15	(8.63)	(16.1)	(9.7)	(3.14)	
67% Visual Feedback	23.4	63.6	166	22.6	4/11
n = 15	(6.05)	(9.19)	(5.89)	(2.13)	
33% Visual Feedback	22.5	63.2	166	22.8	7/8
n = 15	(6.17)	(11.3)	(8.35)	(3.38)	

Data are expressed as the mean (standard deviation). An analysis of differences between groups was performed on the variables listed in the table, but pairwise comparisons revealed no differences. However, the authors noted an uneven distribution of males and females between groups and found that there were no sex differences in the postural control variables analyzed at pre-test. BMI = body mass index.

## Data Availability

The data presented in this study are available upon request from the corresponding author. The data are not publicly available due to privacy or ethical restrictions.

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
