# Peer review of "Effect of Reduced Feedback Frequencies on Motor Learning in a Postural Control Task in Young Adults"

_sensors, 2024, doi:10.3390/s24051404_

Round 1

Reviewer 1 Report

Comments and Suggestions for Authors

MANUSCRIPT TITLE: Effect of reduced feedback frequencies on motor learning in a postural control task in young adults

General Comments

The purpose of this study to verify whether the guidance hypothesis by analysing the effects of different concurrent visual feedback frequencies (i.e., 0%, 33%, 67% and 100%) on performance in a postural control task.

Specific Comments

Abstract

Line 17-18: ² Recent reviews and a meta-analysis concluded that many of the previous studies are statistically underpowered and show possible biases in obtaining further conclusions. ²

This phrase, in my opinion, does not adequately address the study's gap in the abstract. Please check and/or modify.

Introduction

The introduction section does not clearly support the gap in the study and specifically based on specific findings related to the aim of the study.

Suggestion to add a section that focuses on what’s already known.

Procedure

Line 118-119: ²During the second session, subjects performed the retention test.²

Suggestion to add the delayed time between the end of the post test and the retention test.

Line 122-126: ²Both in pre-test, post-test and retention, three trials of the AP task were carried out to evaluate their performance. In addition, in all evaluations, three additional 30-second trials of the task were performed but placing the seesaw in a mediolateral (ML) balance position, to assess the effects of the training on the transfer to another similar postural control task.²

Suggestion to indicate if there is a rest time between trials of both the AP, ML and the two sessions.  

Line 135: ² On the screen was projected an x-y graph, in which the x-axis represented time and the y-axis the CoP position in the AP direction (Figure 1).²

Not clear what mean CoP?

Line 138-140: ²Also, as mentioned above, the experimental groups received different frequencies of feedback: i) group 100% feedback (on all trials), ii) group 67% feedback (on 2 out of 3 trials) and iii) group 33% feedback (on 1 out of 3 trials).²

Suggestion to indicate how reduced feedback conditions was obtained (no-visual condition).

Data analysis

Line 158-160: ² The performance of the tasks (both in ML and AP directions) was assessed using the root-mean-square (RMS) of the signal in the direction of the instability of the seesaw.²

Suggestion to add the formula used to calculate the root-mean-square.

Not clear the outcomes of the calculated variables used in the data analysis.

Results

²Regarding AP, RMS mixed model ANOVA revealed an interaction effect between testing time and feedback group (F6,112 = 3.26; p = 0.005; ƞ2p = 0.15).²

Not clear the reason to only present results of the interaction. Suggestion to add post-hoc analysis.

Discussion

Suggestion to consider the change related to the omitted post-hoc results when rewriting this section.  

"The authors should provide an explanation for the observed performance improvement in the reduced feedback 

Comments on the Quality of English Language

Moderate editing of English language required.

Reviewer 2 Report

Comments and Suggestions for Authors

The authors evaluated the guidance hypothesis in a postural control task by enough statistical power. The results indicated that the use of a reduced frequency of 67% was a better option for improving motor learning than options that offered feedback at a lower frequency, at all trials or not at all. The topic is interesting slightly, but this study lacks novelty. In addition, there are some major issues in the methodology.

Major comments

1.        The authors should show that the ability of static postural control was not different across the four groups before the training because the mean values at Pre-test would be different across the groups in both AP and ML directions in Figure 2.

2.        The readers would suspect the accuracy of COP data when the Wii board inclined.  The authors should show the validation of RMS of COP as the validation of the time of stability and median frequency in the study of Sánchez-Tormo A et al. In addition, the other parameters such as COP area and COP velocity could help the discussion about postural control.

3.        The authors should discuss the neurological mechanisms why the adequate visual feedback frequency 67% but not 100% and 33%, contributes the motor learning.

4.     The postural stability in the ML direction was very sufficient compared to that in the AP direction during the feedback training in Figure 1B, that may be the reason for the no transfer from the postural stability in the AP direction to that in the ML direction.

Reviewer 3 Report

Comments and Suggestions for Authors

In the present work the authors studied the influence of different frequencies feedback on the motor learning by exploring the performance in postural control tasks before and after training. This manuscript provides evidence that using reduced feedback frequencies enhance performance in defined motor activities. Specifically, the study found that 67% feedback frequency generate a better performance than attempts with 100%, 33% and 0% feedback.

The research topic is interesting in the field and a logical continuation of the authors’ previous work. The experimental design is robust and executed correctly. The results showed significant findings about the effect of feedback frequencies on the subjects performance.

On the other hand, I am including below some suggestions to enhance the readability and comprehension of this manuscript. However, I consider the manuscript will need English proofreading. 

Minor points

Abstract:

Line 21-21. “In addition, we tested whether had a transfer in performance on another similar postural control task” replace by In addition, we tested whether there was a transfer in performance to another similar task involving postural control.

Introduction

Line 39. Replace “of himself/herself” by from himself/herself

Line 77-80. The sentence is too large. Please, improve it.

The results describing the performance for the additional task involving a similar postural control (ML) are interesting. However, it is lacking a short description about the plausible underlying mechanisms.

Materials and Methods

The procedure section should be improved. I suggest the authors to include a figure illustrating the protocol temporal course  and rewrite the protocol description in the text.

Line 130-131. Something is lacking in the sentence “However, the experimental groups received concurrent visual feedback during task performance, although each group with a different frequency”. Please, reformulate it

Line 136. The initials CoP is not explained. Although the authors have included references, this concept should be previously described in the current manuscript.

Figure 1. It would be more appropriate to include a figure generated from the obtained waveform instead of a screenshot.

Discussion

Line 246-250. The first sentence in too large. Please, rewrite it

Line 250-251. These results are fully in line with previously published results with postural control task [34] and other different types of feedback and transfers. [35–37]. This sentence does not provide any explanation about the results obtained for postural control task in the ML direction. The references are relevant, but the authors should delve into this topic at the discussion section in order to formulate a plausible explanation about results.

Comments on the Quality of English Language

The manuscript needs English proofreading in order to improve its understanding.

Round 2

Reviewer 1 Report

Comments and Suggestions for Authors

I suggest to have a check on the new added section (English languge).

Comments on the Quality of English Language

I suggest to have a check on the new added section (English languge).

Reviewer 2 Report

Comments and Suggestions for Authors

The manuscript has been improved well. There are some minor issues.

1. The standing posture on the bord such as feet width, foot angle and arm position should be described.

2. Please describe how the "0" point of the CoP was defined. If the point was not correlated with the standing position in the AP direction, the method should be described in the limitation.

3. Figure 2:

- The description of Mean Velocity should be described in the legend.

- The indication of significant differences in Mean Velocity AP and ML should be deleted because of no interaction effects between group and test time.
